# Lignocellulosic Biomass from Grapevines as Raw Material for Particleboard Production

**DOI:** 10.3390/polym14122483

**Published:** 2022-06-18

**Authors:** Radosław Auriga, Alicja Auriga, Piotr Borysiuk, Jacek Wilkowski, Olga Fornalczyk, Ireneusz Ochmian

**Affiliations:** 1Institute of Wood Sciences and Furniture, Warsaw University of Life Sciences—SGGW, ul. Nowoursynowska 159, 02-776 Warsaw, Poland; piotr_borysiuk@sggw.edu.pl (P.B.); jacek_wilkowski@sggw.edu.pl (J.W.); olgafornalczyk@onet.pl (O.F.); 2Faculty of Biotechnology and Animal Husbandry, West Pomeranian University of Technology Szczecin, Janickiego 33, 71-270 Szczecin, Poland; alicja.auriga@zut.edu.pl; 3Faculty of Environmental Management and Agriculture, West Pomeranian University of Technology Szczecin, ul. Słowackiego 17, 71-434 Szczecin, Poland; ireneusz.ochmian@zut.edu.pl

**Keywords:** agricultural waste, biomass, particleboards, wood-based composites, vine pruning

## Abstract

The study aimed to determine the suitability of agricultural lignocellulosic biomass in the form of vine pruning waste for particleboard production. Two variants of particleboards with densities of 650 kg/m^3^ and 550 kg/m^3^ containing a varied amount of vine pruning waste (0, 25, 50 and 100%) were evaluated. The strength (MOR, MOE and IB), thickness swelling and water absorption after immersion in water for 2 and 24 h were tested. The results revealed that vine pruning waste affected the board thickening and reduced strength properties. Boards with a 50% share of waste met the minimum requirements of strength properties specified in the EN 312 standard for boards with a density of 650 kg/m^3^. However, boards with a density of 550 kg/m^3^ entirely made with vine pruning waste met the minimum requirements of strength properties of the EN 16368 standard. Moreover, the pruned material reduced axial forces during drilling, swelling and water absorption.

## 1. Introduction

In recent years, more and more attention has been devoted to the bioeconomy defined as a set of activities aimed at generating economic benefits through the efficient and sustainable use of biological resources. These activities include the production and sale of organic products and bioenergy obtained from the processing of organic matter not intended for human or animal consumption [1].

The most significant sector affecting the bioeconomy is agriculture—the largest producer of organic biomass [2,3]. The assumptions of the bioeconomy contribute to developing innovative ecological solutions aimed at converting plant waste into value-added products such as food, feed, bioproducts and bioenergy [2,4,5]. It is noteworthy that over the last two decades the research on biomass utilization has been mainly investigating power engineering, particularly energy acquisition. However, a significant part of agricultural waste classified as lignocellulosic materials can be also successfully used in other industries.

One of the possible ways of managing lignocellulosic waste from agriculture is its utilization in particleboard technology, which is primarily based on wood feedstock. The volume of produced particleboards in 2020, only in Europe, was 40 million m^3^, and the world’s production was about 96 million m^3^ [6]. Such a high value of the production requires undisrupted availability of wood raw material, which is limited in the market. Additionally, the growing demand for the feedstock corresponds to an increase in the price. Therefore, a possibility of utilizing a lignocellulosic material which is an agricultural waste, might be an important direction for the particleboard industry to find an alternative feedstock.

Agricultural wastes are easily available and commonly found in large quantities. The research on their utilization in the particleboard industry is carried out all over the world. So far, investigations included attempts to produce particleboards from, e.g., Miscanthus stalks [7], wheat straw [8], rice husks [9], kenaf stalks [10], sunflower stalks [11,12], tomato stalks [13], almond shells [14], kiwi prunings [15], apple prunings [16], vine prunings [17,18,19,20], sugar beet pulps [21] and hemp shives [22].

A utilization of pruned material from grapevines seems to be an interesting and noteworthy alternative for the particleboard industry [17,18,19,20]. Ntalos et al. [18] indicated that an increase in the share of vine pruning waste deteriorates mechanical and physical properties in single-layer particleboards. In turn, Yeniocak et al. [19] reported that three-layer particleboards with vine prunings were characterized by better mechanical and physical properties than boards made of pinewood—the best properties exhibited boards consisting of 25% of vine pruning waste.

In 2020 the world’s winegrowing area was over 7.3 mha with 50% of this area located in 5 countries: Spain (13.1%), France (10.9%), China (10.7%), Italy (9.8%) and Turkey (5.9%) [23].

Annual vine pruning is a source of a large amount of biomass. It is estimated that about 5 tonnes of pruned material is produced from one hectare of crop [17,18]. Only in Europe, the total winegrowing area in 2020 exceeded 3.2 mha [24]; is what gives nearly 16 million tonnes of the material per year. Vain pruning waste finds little use as an energy source, but still large amounts of this raw material remain unused. Its utilization in the production of particleboard creates great opportunities and convenes the goals and objectives of bioeconomy.

The aim of this study was to determine whether vine pruning waste can be used as a partial or complete substitute for wood raw material in the production of three-layer particleboards. As part of the research, the mechanical and physical properties of the manufactured particleboards with a standard density of 650 kg/m^3^ and boards with a reduced density of 550 kg/m^3^ were determined.

## 2. Materials and Methods

### 2.1. Raw Materials

Pinewood particles and particles obtained from vine pruning waste were used for the manufacturing of particleboards. Industrial pinewood particles were supplied by a particleboard company, which is located in north-eastern Poland. The particles for face and core layers were sorted under industrial conditions, and their fractional composition is shown in Table 1.

The lignocellulosic biomass was vine shoots of the Regent variety obtained from annual pruning carried out between February and March in 2017 in a non-irrigated 8-year-old vineyard located at the Research Station in Ostoja of the West Pomeranian University of Technology in Szczecin (53°40′ N, 14°45′ E). The vine pruning waste was ground to particles on a laboratory knife cutter. The obtained particles were dried to a moisture content of approx. 4%, and then sorted into particles for core and face layers (Figure 1). The shavings for the core layers passed through a 6 mm sieve and formed a residue on a 2 mm sieve. The particles for the face layers passed through a 2 mm sieve and constituted a residue on a 0.25 mm sieve. The fractional composition of particles for individual layers is illustrated in Figure 1.

### 2.2. Adhesives

Urea-fromaldehyde resin (Silekol 123) was used as a binder and a 10% ammonium sulfate solution was applied as a hardener. Selected properties of the resin are summarized in Table 2. Unit recipe of adhesive was as follows: 50 parts by weight of UF resin, 15.5 parts by weight of water and 1.5 parts by weight of hardener.

### 2.3. Particleboard Manufacturing

As part of the research, three-layer particleboards were produced in two density variants, 550 and 650 kg/m^3^. The boards had dimensions of 320 × 320 mm and thicknesses of 16 mm. The adhesives content of the face layers was 12% and for the core layer 10%. The face layers constituted 35% of the board. The content of vine pruning waste particles in the plates was varied: 25%, 50% and 100%; however, the content of vine pruning material was the same in the face and core layers. The reference boards (0%) were made only with industrial pine particles. Designations of individual variants of produced particleboards are presented in Table 3.

The board pressing process was carried out on a single shelf press at 180 °C and using a maximum unit pressure of 2.5 MPa, pressing factor 18 s/mm (time of pressing 4.48 min). After manufacture, the boards were conditioned under room conditions, i.e., 20 ± 2 °C, 65 ± 5% relative humidity, for at least 7 days.

### 2.4. Mechanical and Physical Properties

Mechanical properties were determined on the INSTRON 3369 universal testing machine (Nortwood, USA). Determination of a given property was performed at least in 10 repetitions for a given type of panel. Determination of static bending strength (MOR) and flexural modulus (MOE) were carried out according to EN 310 [25]. However, the spacing of supports was 280 mm, and the length of tested samples was 300 mm instead of 320 mm as indicated by the standard. The difference resulted from the manufactured board’s dimensions. The load increase was set so that the sample destruction occurred after 60 ± 30 s from the initiation of the assay.

Tensile strength perpendicular to the plane of the board (IB) was estimated following the EN 319 [26] standard. The load increase was the same as in the case of MOR and MOE tests.

The density of manufactured boards was determined according to the EN 323 [27] standard. Additionally, the density profile was assayed in three replications. The 50 × 50 mm samples were analyzed via a GreCon Da-X (X-ray) measuring instrument (Alfeld, Gemany) with an incremental step of 0.02 mm/s.

The thickness swelling after soaked in water for 2 h and 24 h was determined based on the EN 317 [28] standard. The test involved 10 replicates for each variant.

### 2.5. Susceptibility to Drilling

The Busellato Jet 130 CNC machining center (Casadei-Busellato, Thiene, Italy) was used for machinability tests of the particleboards. For the through-hole drilling (throughout the entire thickness of the plate) a new, 8 mm diameter, single-edge, polycrystalline DPI diamond drill of Leitz, GmbH and Co. KG, Stuttgart, German was used. The set parameters of the drilling were as follow: rotational speed 6000 rpm, feed speed 1.2 m/min and feed per revolution 0.2 mm. Additionally, the Fz axial force signals during drilling were recorded using a Kistler 9345A piezoelectric force sensor (Kistler Group, Winterthur, Switzerland). The sampling frequency was 12 kHz. Ten replications were made for each variant. The RMS of axial force signals was evaluated.

### 2.6. Statistical Analysis

Statistical analysis was performed using Statistica 13 (TIBCO Software Inc.). In order to demonstrate the significance of the examined factors’ impact on the properties of the boards, an analysis of variance (α = 0.05) was performed. A comparison of the means was performed by the Tukey test, with a 0.05 significance level.

## 3. Results

The manufactured particleboards were characterized by an average density ranging from 666–669 kg/m^3^ for variants A, B, C and D, and from 563–579 kg/m^3^ for variants E, F, G and H (Table 4).

The difference between the assumed values of density (650 and 550 kg/m^3^) and obtained ones did not exceed 6%. Moreover, the densities within the groups did not differ significantly (Table 3).

Boards made entirely with industrial particles were characterized by significantly lower differences in densities between the face and the core layers, compared to boards containing vine pruning waste (Figure 2). An increase in the difference between densities of the face and the core layers along with the increase in the content of vine pruning waste in the board were observed. These differences in density profiles indicate a differentiation in the mat compaction method [29]. This process has been undoubtedly influenced by the geometry of the compacted particles [30].

The bulk density of mat, as well as the susceptibility to the compaction of mat layers, depend on the particle size [31]. Considering that pruned material from grapevine and wood particles were used for the production of the boards, some discrepancies in the dimensions of the obtained particles should be expected. Similarly, differences in the density of the used materials directly affect the geometry of the produced particles [32]. Another factor that needs to be considered when discussing the board’s compaction is the chemical composition of the feedstock. The content of lignin as a plasticizing substance in the pressing process may affect the thickening process.

Vine pruning wastes have a lower lignin content (24%) [18] than pinewood raw material (approx. 29%) [33]. Particleboards containing pruned material from grapevine were characterized by a lower density compared to conventional particleboards. Therefore, it might be presumed that different compaction levels of manufactured boards will directly affect the strength properties [34,35].

The most crucial factor in determining the static bending strength (MOR) and modulus of elasticity (MOE) was the share of vine pruning waste in the boards, Pc = 46.66% and Pc = 56.60%, respectively (Table 5). When analyzing the data presented in Figure 3, it can be seen that static bending strength deteriorates as the proportion of the pruning waste increases, regardless of the density of the produced particleboards. A similar relationship can be observed for the modulus of elasticity (MOE) (Figure 3).

Findings of Ntalos and Grigoriou [18] corroborate the deterioration of mechanical properties along with the increase in the content of pruned material from grapevines in the case of single-layer particleboards. The observed relationship is analogous for particleboards made with other lignocellulosic agricultural wastes [16].

According to Ntalos et al. 18 and Yeniocak et al. [19] the observed decrease in the mechanical properties of particleboards with an increase in the content of vine prunings may be caused by the presence of small amounts of pith particles (approximately 7%). Pith consists of parenchyma cells, which are softer and shorter than the other cells; therefore, its strength properties are low. In addition, the decrease in the mechanical properties of the particleboards can be due to a more significant number of extracts in the vine prunings than in wood. The content of extractive substances may deteriorate the wettability of the particles and thus affect the particle gluing process [31].

Moreover, the density of manufactured particleboards significantly affected MOR and MOE properties (Table 5). The board’s strength value decreased as the density decreased. It should be noted that the correlation of density with the basic mechanical properties of particleboards, such as MOR and MOE, is consistent with the literature data [34,36,37].

Particleboards with a density of 650 kg/m^3^ entirely made with particles from vine pruning waste did not meet the minimum requirements of the EN 312 [38] standard for general-purpose use in dry conditions (10 N/mm^2^). According to the EN 16368 standard, panels with a density of 550 kg/m^3^ can be considered lightweight boards. In the present study, all boards manufactured with an assumed density of 550 kg/m^3^ met the standards’ requirements for general-purpose lightweight boards LP1 for use in dry conditions (MOR = 3.5 N/mm^2^; MOR = 500 N/mm^2^). Variants with the content of vine pruning waste from 0% to 50% were characterized by the higher strengths than the minimum requirements specified in the EN 16368 standard for general-purpose lightweight boards LP2 for use in dry conditions (MOR—7.0 N/mm^2^; MOR—950 N/mm^2^).

Tensile strength perpendicular to the planes (IB) was significantly affected by both the vine pruning waste content and the density of the boards. Interestingly, the Pc value of vine pruning waste share was considerably lower than the error Pc value which indicates that untested factors had a greater influence on the parameter. Furthermore, the analysis of the IB results (Figure 3) does not allow to define a clear relationship between the vine pruning content and IB. The parameters’ value was determined by the density of manufactured particleboards, Pc = 66% (Table 5). The observed correlation of IB with the density corresponds to reports of other authors who were studying the use of agricultural waste in the production of particleboards [18,36].

The share of vine pruning waste had a positive effect on swelling thickness and water absorption. Analyzing the results presented in Figure 4, it can be stated that the increase in the share of vine pruning waste in the particleboard decreased swelling and water absorption of the boards. Statistical analysis of the obtained results showed that both the vine pruning share and the density of the boards have a significant effect on swelling thickness and water absorption. It should be noted that in the case of swelling in thickness, the dominant factor was the share of vine pruning waste, which corroborated high values of the contribution percentage; Pc = 75% after 2 h, and Pc = 47% after 24 h of being soaked in water (Table 6).

Some researchers have reported that the value of thickness swelling rises along the increase in the share of the alternative lignocellulosic material originating from agriculture in the board [15,39]. In contrast, Pirayesh and Khazaeian [14] found that the increased content of alternative lignocellulosic material decreases the swelling value of the board. These results corroborate with the observations of Yeniocak et al. [19] of a decrease in thickness swelling properties of boards consisting of vine pruning waste. The reason for this may be a larger content of extractives in the particles of vine prunings than in wood, which may affect the water-repellent properties of the boards produced. Differences in swelling or absorbability values can result from the properties of the raw material used for the particleboard manufacture, and from the production parameters of these boards [10]. In addition, the value of swelling in thickness depends on the density of the manufactured boards. Generally, the increase in density of manufactured particleboard causes an increase in thickness swelling [39].

Machinability is one of the key properties of wood-based panels. The easy machining of materials determines the possibility of giving the right shapes to processed composites for a given type of application. Analyzing the axial forces for drilling shown in Figure 5, it can be seen that the increase in the content of vine pruning waste caused a slight decrease in the value of the axial force for drilling. It should be noted, however, that the decrease in the parameter, in this case, was not statistically significant. The dependence of axial force during drilling on the density of the produced particleboard is much more visible. The decrease in density corresponds to the decrease in the axial force. This relationship corroborates the study of Podziewski et al. [40].

## 4. Conclusions

Vine pruning waste affected the mat thickening process and reduced strength properties. Use of a 50% share of vine pruning waste enables the production of boards meeting the minimum requirements of strength properties specified in the EN 312 standard for boards with a density of 650 kg/m^3^. Boards with a density of 550 kg/m^3^ entirely made with vine pruning waste met the minimum requirements of strength properties specified in the standard EN 16368. The addition of pruned material reduced axial forces during drilling, as well as, swelling and water absorption after immersion in water for 2 and 24 h.

## Figures and Tables

**Figure 1 polymers-14-02483-f001:**
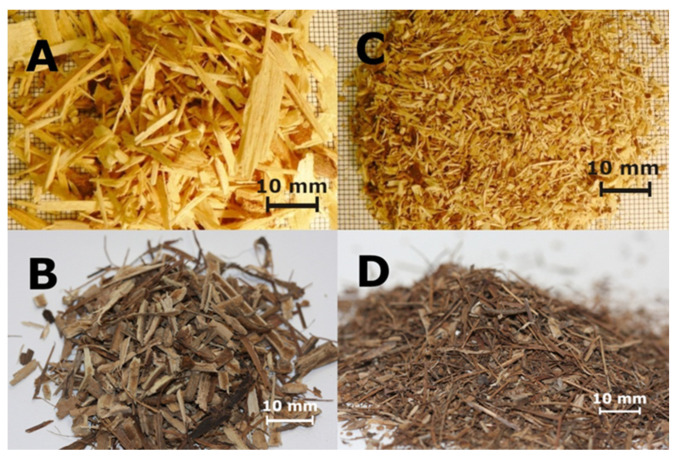
The particles of vine pruning waste: particles for the core layer are (**A**) pinewood and (**B**) vine pruning; particles for the face layers are (**C**) pinewood and (**D**) vine pruning.

**Figure 2 polymers-14-02483-f002:**
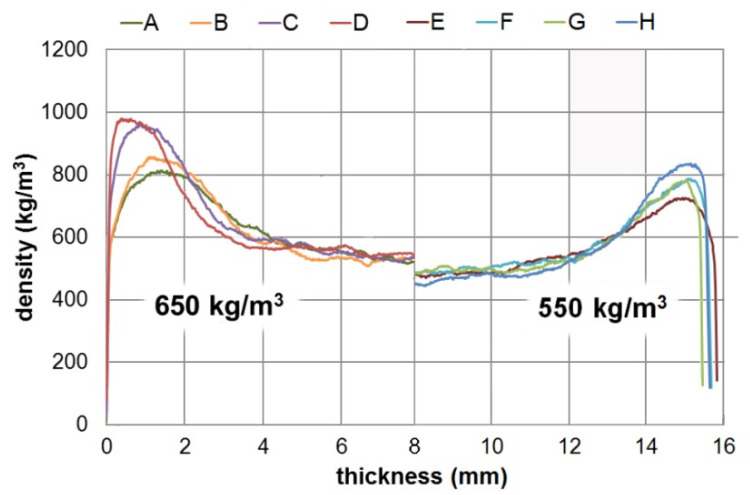
Density profile of manufactured particleboards.

**Figure 3 polymers-14-02483-f003:**
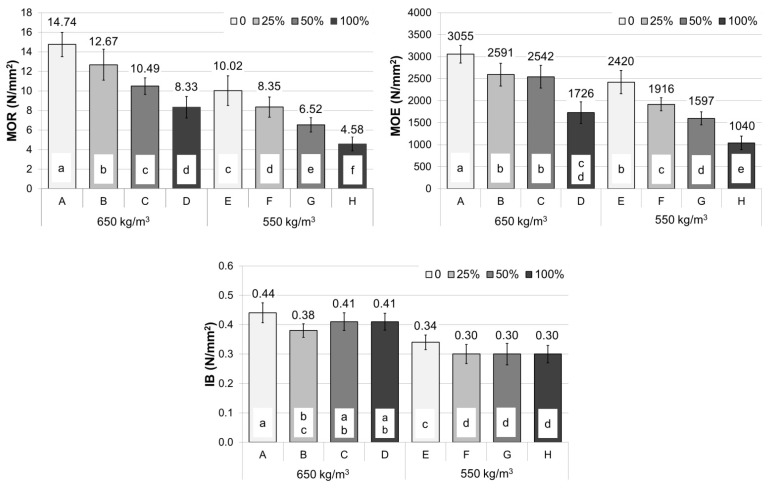
Mechanical properties of particleboards, a–f correspond to the homogeneous groups by Tukey test (α = 0.05).

**Figure 4 polymers-14-02483-f004:**
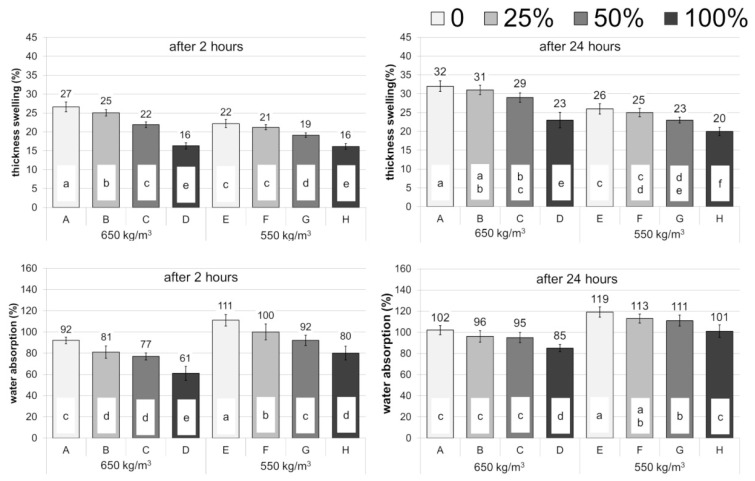
Physical properties of particleboards, a–f correspond to the homogeneous groups by Tukey test (α = 0.05).

**Figure 5 polymers-14-02483-f005:**
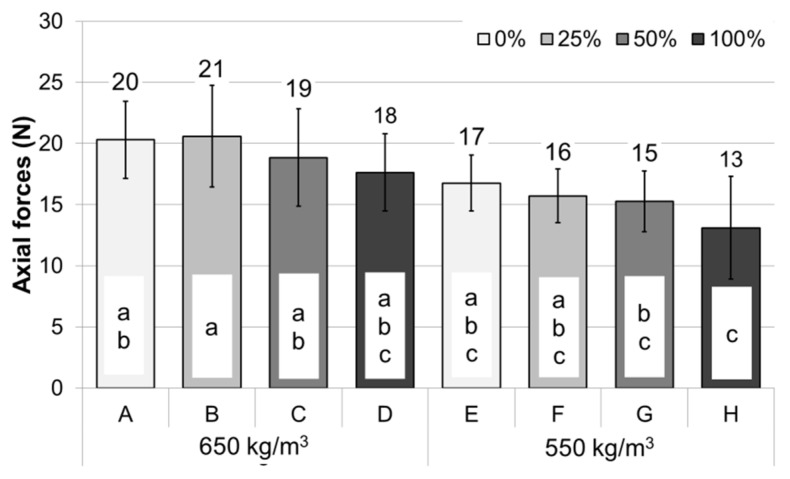
Axial forces during drilling, a–c correspond to the homogeneous groups by Tukey test (α = 0.05).

**Table 1 polymers-14-02483-t001:** Fractional composition of used raw materials.

Fraction (mm)	Industrial Wood Particles		Vine Pruning Waste Particles	
	Core Layers	Face Layer	Core Layers	Face Layer
6.00	13.0	-	26.6	-
4.00	19.0	-	43.7	-
2.00	51.0	0.6	18.4	-
1.25	13.0	14.3	6.1	17.6
0.63	3.6	55.5	3.2	39.4
0.49	02	11.4	0.6	7.6
0.315	0.1	7.8	0.6	19.0
below 0.315	0.1	10.4	0.8	16.4

**Table 2 polymers-14-02483-t002:** Properties of UF resin (Silekol 123).

Characteristic	Value
dry mass	67.0%
pH	8.0
dynamic viscosity	500 mPas

**Table 3 polymers-14-02483-t003:** Variants of manufactured boards.

Variant	Density (kg/m^3^)	Share of Vine Pruning Particles (%)
A	650	0
B	650	25
C	650	50
D	650	100
E	550	0
F	550	25
G	550	50
H	550	100

**Table 4 polymers-14-02483-t004:** Average density of manufactured particleboards.

		Variant							
		A	B	C	D	E	F	G	H
Density (kg/m^3^)	Average	669	668	666	666	579	569	568	563
	±Std. Dev	38	34	38	37	35	34	32	38

**Table 5 polymers-14-02483-t005:** Statistical significance of factor influences on mechanical properties of particleboards.

	MOR		MOE		IB	
	p	Pc (%)	p	Pc (%)	p	Pc (%)
share	0.000	46.66	0.000	56.60	0.000	10.07
density	0.000	41.84	0.000	32.42	0.000	66.18
share x density	0.556	0.32	0.104	0.90	0.206	1.45
error		11.17		10.08		22.30

p—probability of error; Pc—percentage of contribution; x – interaction between factors.

**Table 6 polymers-14-02483-t006:** Statistical significance of factor influences on physical properties of particleboards.

	TS2H		TS24H		WA2H		WA24H	
	p	Pc (%)	p	Pc (%)	p	Pc (%)	p	Pc (%)
share	0.000	74.91	0.000	47.57	0.000	52.42	0.000	30.12
density	0.000	14.76	0.000	39.67	0.000	35.35	0.000	53.33
share x density	0.000	5.03	0.001	2.53	0.542	0.36	0.951	0.08
error		5.29		10.23		11.87		16.46

p—probability of error; Pc—percentage of contribution; x—interaction between factors.

## Data Availability

Auriga, Radoslaw; Auriga, Alicja; Borysiuk, Piotr; Wilkowski, Jacek; Olga, Fornalczyk; Ochmian, Ireneusz (2022), “Lignocellulosic biomass from grapevines as raw material for particleboard production”, Mendeley Data, V1, doi: 10.17632/ch2hcr9zsm.1.

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
