# Peer review of "Lignocellulosic Biomass from Grapevines as Raw Material for Particleboard Production"

_polymers, 2022, doi:10.3390/polym14122483_

Round 1

Reviewer 1 Report

This paper investigates the preparation and properties of particleboard using vine pruning waste. Although the research is interesting, several issues to be addressed:

  1. At the end of the Introduction, the authors should elaborate the aim of their study as well as the novelty compared to previous studies [17–20].
  2. Line 231, “The share of vine pruning waste had a positive effect on swelling thickness and water absorption after the soaked in water.” It is unclear.
  3. For all physical and mechanical properties, the authors just describe the details of the result. Some impact factors accounting for the changes are discussed, involving density, content of the agricultural waste, and process. But how do these impact factors work? Why the addition of vine pruning waste deteriorate the MOR and decrease the water absorption?
  4. I recommend the authors add the comparison between wood particle and vine pruning waste at the aspect of physical and chemical information. In addition, the microstructure of the particle board needs to be evaluated to observe the changes and interface.
  5. The quality of all figures need to be improved.

Author Response

We thank the Reviewers for their professional comments on the manuscript. The manuscript has been revised according to the Reviewers' comments and suggestions. All the sentences pointed out by the Reviewers have been rephrased and clarified. Grammar and spelling correction as well as vocabulary standardization has been made throughout all manuscript. There also has been added novelty of the work and aim in section “Introduction”. Quality of Figures has been improved as well. The responses to further, substantive comments of the Reviewers are outlined below.

Review 1

For all physical and mechanical properties, the authors just describe the details of the result. Some impact factors accounting for the changes are discussed, involving density, content of the agricultural waste, and process. But how do these impact factors work? Why the addition of vine pruning waste deteriorate the MOR and decrease the water absorption?

The text contains numerous references to the literature on the subject, comparing the obtained results with the literature data. The final mechanical properties of the manufactured particleboards are influenced by many factors, not only the material ones but also the sizing process, auxiliaries' presence, and the technological process itself. Therefore, in the authors' opinion, presenting the obtained strength results and discussing them in terms of literature data is sufficient to determine the usefulness of vine pruning in chipboard technology.

The question of the impact of the use of vine pruning on the swelling in thickness and water absorption is included in the paragraph (lines 256-264), a more detailed analysis referring directly to the chemical composition and structure of the raw material used for the production of particleboard and the impact of the technological process itself on these properties is a pervasive topic and can be the basis of many articles. The work focuses primarily on the applicative nature of the research.

I recommend the authors add the comparison between wood particle and vine pruning waste at the aspect of physical and chemical information. In addition, the microstructure of the particle board needs to be evaluated to observe the changes and interface.

Undoubtedly, a complete and thorough analysis of the chemical composition as well as the determination of physical properties would enrich the scientific article, nevertheless, the article aims to determine the impact of vine pruning waste on the final product, which is particleboard. The authors approached the research from the application point of view because the chemical composition of the raw material is not tested in the process of manufacturing chipboards on an industrial scale.

Reviewer 2 Report

Comments and suggestions:

- Introduction: This section of the article should be rewritten, with more focusing on the recent advances in the use of vine prunings for the manufacture of particleboards or other type of wood-based composites. Moreover, it should be important and useful to provide a data about chemical composition and physical and mechanical properties of this biomass as well as comparison them with the conventional wood raw material. Please provide also a recent information about the available volume of such raw material (biomass) in EU countries.

- From this analysis, it should follow what needs to be done to attract this kind of raw material for the manufacture of particleboard.

- Please clear describe the aim and objectives of the study at the end of Introduction section.

- Please improve the quality of Figures 2, 3, 4 and 5.

- What were the densities of the pinewood and vine pruning particles?

- Results: This section requires more discussion, especially regarding the comparison of the results of this study with the results obtained by other authors, which used vine prunings for particleboard production.

- Conclusion: Please provide some perspectives for further research and recommendation for practical use of such raw material for particleboard production. 

Author Response

We thank the Reviewers for their professional comments on the manuscript. The manuscript has been revised according to the Reviewers' comments and suggestions. All the sentences pointed out by the Reviewers have been rephrased and clarified. Grammar and spelling correction as well as vocabulary standardization has been made throughout all manuscript. There also has been added novelty of the work and aim in section “Introduction”. Quality of Figures has been improved as well. The responses to further, substantive comments of the Reviewers are outlined below.

Review 2

Introduction: This section of the article should be rewritten, with more focusing on the recent advances in the use of vine prunings for the manufacture of particleboards or other type of wood-based composites. Moreover, it should be important and useful to provide a data about chemical composition and physical and mechanical properties of this biomass as well as comparison them with the conventional wood raw material. Please provide also a recent information about the available volume of such raw material (biomass) in EU countries.

Vine pruning waste is not utilised by the particleboard industry yet, and research into its use includes a small number of works, primarily by Ntalos and Grigoriou, Yeniocak et al. which the authors cited in the Results section. According to the authors, an important criterion for the applicability of vine pruning waste is not only its suitability for particleboard production but also the availability of the raw material, which in the case of many of the raw materials tested is a key problem. Updated information regarding both the production of particleboard and the area under vineyards has been introduced in the text of the manuscript.

From this analysis, it should follow what needs to be done to attract this kind of raw material for the manufacture of particleboard.

The purpose of the article was to show that the production of particleboard with vine prunings is possible, and the vine pruning itself, considered mainly waste, is a valuable resource. This goal was achieved because the study showed that it is possible to produce vortex plates with the use of vine pruning, which will meet the requirements of these standards in terms of mechanical and physical properties. However, the use of this raw material is limited by its availability and logistics, as well as the seasonality of this raw material.

What were the densities of the pinewood and vine pruning particles?

Unfortunately, this was not specified.

Results: This section requires more discussion, especially regarding the comparison of the results of this study with the results obtained by other authors, which used vine prunings for particleboard production.

References to previous research on the use of vine pruning and other agricultural raw materials in the production of particleboards are found in the text of the Results chapter, e.g.:

Line 207  -210 Findings of Ntalos and Grigoriou [18] corroborate the deterioration of mechanical properties along with the increase in the content of pruned material from grapevines in case of single-layer particleboards. The observed relationship is analogous for par-ti-cleboards made with other lignocellulosic agricultural waste [16].

Line 231 – 233 The observed correlation of IB with the density corresponds to reports of other authors who were studying the use of agricultural residues in the production of particleboard [18,33].

Reviewer 3 Report

The submitted article “Lignocellulosic biomass from grapevines as raw material for particleboard production” deals with the application of lignocellulosic material obtained from grapevines for the production of particle boards. Considering the amount of waste expelled out of grape production for wine making, the topic look relevant for consideration for possible publication. However, there is scope of improvement for this manuscript and the suggestions are as follow:

Abstract:

Line 14 : The sentence is confusing, please rephrase.

Line 19: Is it mat thickness or board thickness? Please clarify

Key words: Is it agricultural residue or waste? Because it is referred as waste in rest of the manuscript.

Introduction:

Line 37-39: Please rephrase the sentence. “However, a significant part of agricultural waste classified as ligno-cellulosic materials can be also successfully used in other industries.” Instead of “However, a significant part of ligno-cellulosic materials classified as agricultural waste can be also successfully used in other industries.”

Line 45 to 47: Please rephrase the sentence. “Therefore, a possibility of utilization a lignocellulosic material being an agricultural waste might be an important direction for finding an alternative feedstock for the particleboard industry.” Instead write “Therefore, a possibility of utilization an agricultural waste in the form of lignocellulosic material might be an important direction for finding an alternative feedstock for the particleboard industry.”

There are several instances where such sentences have been used so, please change the same throughout the manuscript.

Line 48: “Agricultural residues” kindly use same term throughout the manuscript either residue or waste.

Clearly write Novelty of the work.

Aim and objectives should be written.

Materials and Methods

Line 69: Manufacturing not manufacture

Line 70: Particleboard manufacturer

Table 2: The values were measured or derived from somewhere. If measured then provide the procedures for the same and not then provide the reference.

Line 110: If the pressure is 2.5 MPa then how the density is varied?

Line 114: Heading is ‘Physical and Mechanical Properties’ then write physical properties first then mechanical properties else change heading to ‘Mechanical and Physical Properties’

Line 118: Write the author’s name for the reference.

Line 124: Write the author’s name for the reference and follow same for rest of the manuscript if you have similar cases further.

Results and Discussion

It will be easy to understand if subheadings are given in results and discussion section.

Line 149 to 154: Make single para

Table 4: Write it as Value ± Std. Dev.

Line 179 to 181: There is no analysis of composition of lignocellulosic material.

Table 5: What is difference between probability of error and error?

Line 231: Is the water absorption is desirable?

General Comment: Language need to be improved and grammatical errors need to be corrected.

References: Strictly follow the MDPI layout. Refer any article recently published and revise the references.

Author Response

We thank the Reviewers for their professional comments on the manuscript. The manuscript has been revised according to the Reviewers' comments and suggestions. All the sentences pointed out by the Reviewers have been rephrased and clarified. Grammar and spelling correction as well as vocabulary standardization has been made throughout all manuscript. There also has been added novelty of the work and aim in section “Introduction”. Quality of Figures has been improved as well. The responses to further, substantive comments of the Reviewers are outlined below.

Review 3

Table 2: The values were measured or derived from somewhere. If measured then provide the procedures for the same and not then provide the reference.

The data in table 2 were not measured during the investigation, these are data provided by the producers of resin, therefore a reference.

If the pressure is 2.5 MPa then how the density is varied?

2.5 MPa is the unit pressure used during the pressing of the matrices. The matrix is compacted at a given pressure to a given thickness. The unit pressing pressure does not affect the average density of the produced board, but it does affect the density profile presented in the article.

It will be easy to understand if subheadings are given in results and discussion section.

In the authors' opinion, the subheadings in this section are redundant and would unnecessarily introduce artificial divisions in the analysis of results and discussion.

Line 179 to 181: There is no analysis of composition of lignocellulosic material.

In this study, no chemical analysis of the raw material was performed. However, the authors cited the chemical composition of the tested material and counted it as a possible factor influencing the differences in the obtained results.

Table 5: What is difference between probability of error and error?

Probability of error is the result of the test and allows to determine whether the null hypothesis that a given factor is statistically significant is true or not. With the value of probability of error at the level of p <0.05, we can conclude that a given factor is statistically significant. Error means "the variability within the groups" or "unexplained random error." It allows to assess whether the examined factors are the main factors influencing the tested value.

Line 231: Is the water absorption is desirable?

Water absorption is an undesirable feature in the case of particle boards which are widely used in the furniture and construction industries.

Reviewer 4 Report

Journal of Polymers

Technical, grammatical, and common mistakes are as follows

Reviewer Comments

  • Type of the Paper (Article, Review, Communication, etc.)? what’s meant by this? Remove it
  • Write keywords in alphabetical order.
  • The introduction is too short. Revise it with more literature data.
  • Urea-fromaldehyde resin (Silekol 123). Mentioned their purity and manufacturer.
  • Table 3. Explain it with more explanation and compare it with the latest research data.
  • Figure 2, and 4, is not clear. Revise it. Use TIFF 600-dimension format.
  • For %, °C, wt % and for Figure, etc., follow the same format throughout the manuscript.
  • Tensile strength perpendicular to the planes (IB) was significantly affected by both the vine pruning waste content and the density of the boards. [Give a reason with proper reference citations.
  • The conclusions are too short, make them with more explanation.
  • Put all the references through endnote software.

Cite the following references;

  • doi:10.1007/s10924-021-02142-1
  • https://doi.org/10.1007/s10924-021-02337-6

Author Response

We thank the Reviewers for their professional comments on the manuscript. The manuscript has been revised according to the Reviewers' comments and suggestions. All the sentences pointed out by the Reviewers have been rephrased and clarified. Grammar and spelling correction as well as vocabulary standardization has been made throughout all manuscript. There also has been added novelty of the work and aim in section “Introduction”. Quality of Figures has been improved as well. The responses to further, substantive comments of the Reviewers are outlined below.

Reviewer 4

The introduction is too short. Revise it with more literature data.

Several new literature sources were added to the article:

  1. Zvirgzds, K.; Kirilovs, E.; Kukle, S.; Gross, U. Production of Particleboard Using Various Particle Size Hemp Shives as Filler. Materials (Basel)., 2022, 15 (3), 1–19. https://doi.org/10.3390/ma15030886.
  2. Boruszewski, P.; Borysiuk, P.; Maminski, M.; Czechowska, J. Mat Compression Measurements during Low-Density Particleboard Manufacturing. BioResources, 2016, 11 (3), 6909–6919. https://doi.org/10.15376/biores.11.3.6909-6919.
  3. Niemz, P.; Sandberg, D. Critical Wood-Particle Properties in the Production of Particleboard. Wood Mater. Sci. Eng., 2022, 0 (0), 1–2. https://doi.org/10.1080/17480272.2022.2054726.

Urea-fromaldehyde resin (Silekol 123). Mentioned their purity and manufacturer

The name Silekol 123 defines a sufficient adhesive resin, both in terms of the manufacturer and the specific product. In addition, table 2 shows the basic properties of this resin.

Tensile strength perpendicular to the planes (IB) was significantly affected by both the vine pruning waste content and the density of the boards. Give a reason with proper reference citations.

The IB value might have been influenced by both the pressing process during production and the type of raw material used as well as many other factors. Therefore, the authors referred to the findings of researchers whose results were similar and addressed the IB analysis in more detail.

The conclusions are too short, make them with more explanation.

In the opinion of the authors, the conclusion section should consist of a synthetic presentation of the outcomes of the work. On the other hand, the explanation of the observed phenomena is included in the results analysis and their discussion.

Round 2

Reviewer 1 Report

Although the authors revised the manuscript, the queries I mentioned still not solved.

1. The authors cited Ref. 17-20 which are related to the particleboard using vine pruning. Compared to those studies, what is the novelty of the current study?

2. The chemical information of the vine pruning is more important for the reader of Polymers than the properties of the resulting products. In other words, we want to see the mechanism of the properties changes and why the substitution of vine pruning for wood particle is feasible.

Author Response

Response to comment 1:
Cited studies were conducted on particleboards with a density of 700 kg/m3, while the industry produces 550 and 650 kg/m3 particleboards. It is due to the sustainability of the use of natural feedstock in particleboard production. Nevertheless, authors have included in the Introduction some additional information about cited research.

Response to comment 2:
Thank you for this comment. The chemical composition of the used feedstock in the study was not the priority of the research. Therefore, a detailed chemical analysis wasn’t conducted. However, the aspects of chemical composition were cited in the text. Authors agree that chemical information on used material could enrich the presented work, but on the other hand, in our opinion, that topic is very wide and complex and deserves separate research. However, an additional explanation of  mechanical and physical properties has been added to the section “Results”.

Reviewer 2 Report

The manuscript has been revised according to my comments and suggestions, and it has been sufficiently improved for publication in Polymers. 

Author Response

Thank you for your professional comments and time.

Reviewer 3 Report

Thanks for revising the article. I hope the article will be accepted for possible publication.

Author Response

(The authors gave the same response as above.)

Round 3

Reviewer 1 Report

I do not have queries.